# The Impact of Substituting Chalk with Fly Ash in Formulating a Two-Component Polyurethane Adhesive on Its Physicochemical and Mechanical Properties

**DOI:** 10.3390/ma18153591

**Published:** 2025-07-30

**Authors:** Edyta Pęczek, Renata Pamuła, Żaneta Ciastowicz, Paweł Telega, Łukasz Bobak, Andrzej Białowiec

**Affiliations:** 1Department of Applied Bioeconomy, Wrocław University of Environmental and Life Sciences, 37a Chełmońskiego Street, 51-630 Wroclaw, Poland; zaneta.ciastowicz@upwr.edu.pl (Ż.C.); pawel.telega@upwr.edu.pl (P.T.); 2Selena Industrial Technologies Sp. z o.o., Pieszycka 3, 58-200 Dzierżoniów, Poland; renata.pamula@selena.com; 3Department of Functional Food Products Development, Wrocław University of Environmental and Life Sciences, 51-630 Wroclaw, Poland; lukasz.bobak@upwr.edu.pl

**Keywords:** two-component polyurethane adhesive, fly ash, chalk, filler, circular economy principles

## Abstract

This study aimed to evaluate the effect of replacing chalk with fly ash in a two-component polyurethane (2C PU) adhesive on its physicochemical, mechanical, and environmental properties, as a practical application of circular economy principles. Six adhesive formulations were prepared, each containing a chalk-to-fly ash ratio as a filler. The study evaluated rheological, mechanical, thermal, and environmental parameters. Mechanical tests confirmed cohesive failure within the bonded material, indicating that the bond strength at the adhesive–substrate interface exceeded the internal strength of the substrate. The highest contaminant elution levels recorded were 0.62 mg/kg for molybdenum and 0.20 mg/kg for selenium, which represent only 6.2% and 40% of the regulatory limits, respectively. Dissolved organic carbon (DOC) and total dissolved solids (TDS) did not exceed 340 mg/kg and 4260 mg/kg, respectively. GC-MS analysis did not reveal the presence of prominent volatile organic compound emissions. Initial screening suggests possible compatibility with low-emission certification schemes (e.g., A+, AgBB, EMICODE^®^), though confirmation requires further quantitative testing. The results demonstrate that fly ash can be an effective substitute for chalk in polyurethane adhesives, ensuring environmental compliance and maintaining functional performance while supporting the principles of the circular economy.

## 1. Introduction

Two-component polyurethane (2C PU) adhesives are widely used across various industries due to their excellent properties, such as impact resistance, chemical resistance, hardness, abrasion resistance, thermal stability, and the ability to bond different materials, including metals, plastics, and wood. These adhesives consist of two components, polyol and isocyanate, which undergo a chemical reaction upon mixing, resulting in the formation of a durable bond [1,2,3,4].

One of the key advantages of 2C PU adhesives is their customizability, as their properties can be precisely adjusted by modifying the chemical composition of both components [5,6]. To enhance performance and reduce production costs, various fillers such as talc, barium sulfate, cement, or quartz powder are commonly added to the adhesive formulation [7,8,9]. Among them, chalk (CaCO_3_) is the most frequently used due to its rheological and mechanical properties, making it widely applied in both PU foams and adhesives. It acts as a stabilizer, enhances functional properties, and reduces production costs and raw material consumption. Due to its widespread use, chalk serves as a reference point when evaluating alternative fillers, including those derived from industrial waste [10,11,12,13,14,15]. However, due to growing environmental concerns and the need for cost optimization, there is increasing interest in replacing traditional fillers with industrial waste materials that offer comparable properties. In light of global sustainability and regulatory challenges—such as the European Green Deal and the Energy Performance of Buildings Directive (EPBD)—alternative raw materials are being actively explored [16,17,18]. In recent years, there has been a dynamic increase in research on the use of industrial waste as additives in polyurethane (PU) materials. Previous studies have primarily focused on PU foams, composites, and thermoplastic polyurethanes (TPUs), while the topic of polyurethane adhesives—particularly two-component (2C PU) systems—remains insufficiently explored in the scientific literature [19]. At the same time, numerous studies demonstrate that selected waste materials can serve as effective fillers, positively influencing the mechanical, physicochemical, and environmental performance of PU systems.

For example, fly ash has been shown to improve compressive strength, fire resistance, and thermal insulation of rigid PU foams [20,21,22,23], while sunflower husk ash contributes to dimensional stability and biodegradability [24]. Coconut fibers increase bending and shear resistance [25,26], and recycled tire rubber (GTR) enhances elasticity and both acoustic and thermal properties [27,28]. Waste glass ash improves sound absorption performance [29], while fish-derived collagen increases tear resistance and aging stability [30]. Hazelnut shells modified with hydrotalcite improve compressive and bending strength, as well as flame resistance [31]. Compositions based on lignin, rice husks, coconut, and eggshells improve mechanical strength and durability while incorporating a high proportion of bio-based or waste components [32]. Biopolymers such as keratin, chitin, zein, and gelatin allow the synthesis of non-isocyanate polyurethane (NIPU) systems with high renewable content [33]. Additionally, studies on particleboards containing waste plastics emphasize the broader shift toward circular economy practices by demonstrating that industrial plastic waste can successfully replace virgin raw materials while maintaining or improving mechanical and physical properties [34].

Such examples highlight the potential of various waste streams to act as functional fillers that improve mechanical and functional properties. They can also help optimize production costs and reduce the use of primary raw materials. Fly ash, one of the most common industrial wastes produced worldwide, is estimated to have a global production range of 400 to over 750 million tons annually, depending on sources and data collection methods [35]. This substantial scale of production makes fly ash an easily accessible secondary raw material, drawing increasing attention in the context of sustainable material development.

Although fly ash is already widely used in cement and concrete production, its potential in polymer materials, such as polyurethane foams (PU), thermoplastic polyurethanes (TPU), and composites is increasingly being explored. However, very few studies have investigated its application in polyurethane adhesives—and even fewer in two-component (2C PU) adhesive systems. So far, this topic has not been widely addressed in the scientific literature in the context of two-component polyurethane adhesives. A recent mini-review on polyurethane additives indicates that studies on waste fillers primarily focus on foams and composites, while adhesives are rarely examined in this regard [19]. This research, therefore, fills a clear knowledge gap by assessing the mechanical, thermal, and environmental effects of this substitution in adhesive systems with industrial relevance.

Therefore, further research is needed to assess how incorporating this waste into polyurethane adhesive formulations could impact their mechanical and thermal properties and help reduce the use of primary mineral raw materials.

In response to this research gap, the present study investigates the replacement of chalk with fly ash in a two-component polyurethane adhesive formulation. Four adhesive variants were developed, containing 100%, 75%, 50%, and 10% fly ash, with the remaining part composed of chalk. The study assessed key performance parameters, including density, open time, viscosity of individual components and the mixture, thermal conductivity, Shore hardness, tensile strength, elongation at break, stress at break, and heat of combustion. Additionally, environmental factors like contaminants’ leachability and VOC emission were studied. However, the present study focuses on short-term thermal and mechanical properties. Long-term performance aspects such as aging resistance, humidity cycling, or fire resistance were beyond the scope of this work and should be explored in future studies.

The obtained results will provide valuable insights into the feasibility of recycling fly ash as a cost-effective and sustainable alternative to chalk, assessing its impact on both the physicochemical and mechanical properties of polyurethane adhesives. In the long term, this research supports the development of more sustainable adhesive technologies, aligning with circular economy principles and contributing to innovative material solutions in the chemical industry.

## 2. Materials and Methods

The raw materials used for the synthesis included fly ash supplied by Veolia EKOZEC Sp. z o.o. (Poznań, Poland); a triglyceride of plant-based fatty acids, primarily hydroxyoeleic acid, from Standard Sp. z o.o. (Lublin, Poland); short-chain polyol (ethane-1,2-diol) from Brenntag Polska Sp. z o.o. (Kędzierzyn-Koźle, Poland); polymeric methylene diphenyl diisocyanate (p-MDI) from BorsodChem (Kazincbarcika, Hungary); an air-release additive from Byk (Wesel, Germany); and a drying agent (crystalline aluminosilicate) from HSH Chemie Sp. z o.o. (Warsaw, Poland).

All raw materials were tightly sealed and stored for 7 days under standard laboratory conditions (T = 23 ± 1 °C, RH = 50 ± 5%) to minimize the influence of external factors on their properties. The sample preparation was carried out under the same controlled conditions. Each measurement of density, open time, viscosity, tensile stress–strain properties, tensile strength perpendicular to faces, thermal conductivity, Shore hardness A, and high heating value was performed ten times.

Before conducting the measurements of thermal conductivity, Shore hardness A, tensile stress–strain properties, and high heating value, the test specimens were properly prepared and conditioned. Component A was mixed with Component B in a 100:20 ratio. The samples were then left to fully cure for 7 days under controlled conditions of T = 23 ± 1 °C and RH = 50 ± 5% to ensure complete crosslinking.

The density measurement was carried out using a metal pycnometer in accordance with EN ISO 2811-1:2023 (POL-ZAF, Wroclaw, Poland) [36]. The procedure involved filling the calibrated device with the test liquid, measuring its weight, and calculating the density based on the weight difference. The measurement was performed using a Quintix^®^ 6102-1CEU balance (Sartorius AG, Göttingen, Germany).

The open time was determined by manually mixing Component A with Component B, measuring 200 ± 0.1 g of Component A and adding 40 ± 0.1 g of Component B. The sample was then placed under a mechanical stirrer, and both the stirrer and stopwatch were started simultaneously. The mixing process lasted for 40 s until a homogeneous mixture was obtained. The open time was defined as the moment when the mixture lost its ability to flow. The state of the mixture was checked every 5 min starting from the 50th minute.

Viscosity measurements were conducted using the Brookfield CAP 2000+ viscometer by EN ISO 3219-2:2021 (AMETEK Brookfield, Middleboro, MA, USA), utilizing a cone-plate rheometer setup with spindle S05 [37]. For Component A, viscosity was measured at temperatures of 10, 20, 30, 40, 50, 60, and 70 °C. The spindle rotational speed was set to 10 rpm at 10 °C, while for all other temperatures, measurements were performed at 500 rpm.

The viscosity of the mixture, after combining Component A with Component B, was measured in the temperature range of 10 to 50 °C, following the same rotational speed scheme as for Component A. Measurements at 60 °C and 70 °C were omitted due to the viscometer’s excessive stabilization time, which led to unreliable results.

Thermal conductivity was measured according to EN 12667:2001standard [38] using an Isomet 2114 (TYP IPS 1100) analyzer (Applied Precision, Prague, Czech Republic). The test involved placing the samples in controlled temperature conditions and applying the direct thermal conductivity measurement method. The measurement pressed precisely against the surface of the sample to ensure proper thermal contact. The device then measured the thermal conductivity by analyzing the heat flow within the sample.

Shore A hardness was measured using a Manual Shore Test Stand TI-AC (Sauter GmbH, Balingen, Germany), in accordance with EN ISO 868:2003 [39]. The sample was placed on a stable surface, and the durometer was applied perpendicularly to the tested surface. A test force of 10 N was used to ensure uniform contact between the indenter and the sample.

The determination of high heating value (HHV) was carried out following PN-ISO 1928:2020 [40] using laboratory equipment consisting of an IKA C200 calorimeter, IKA C248 oxygen station, and C5010 calorimetric bomb at 17–25 °C (IKA-Werke GmbH & Co. KG, Staufen, Germany), 30 bar pressure.

The tensile strength perpendicular to faces was measured following PN-EN 1607:2013 [41]. The tests were conducted using an Instron 3367 testing machine (Instron, Norwood, MA, USA). The specimens consisted of galvanized steel + XPS and aluminum + XPS bonds. The failure type was classified according to EN ISO 10365:2022 [42].

The tensile stress–strain properties were determined following ISO 37:2024 [43] using a Zwick Roell Z2.5 testing machine (ZwickRoell GmbH & Co. KG, Ulm, Germany). The nominal crosshead speed was set at 200 mm/min, ensuring controlled elongation of the specimens during testing. A preload force of 0.1 N was applied.

The leachability test was conducted following the PN-EN 12457-4:2006 standard [44], and the obtained water extracts were analyzed for pollutant indicators following the PN-EN ISO/IEC 17025:2018-02 standard [45]. The procedure involved a single-step extraction of water-soluble components under static conditions. The total extraction time was 24 h, comprising 6 h of intensive stirring followed by 18 h of settling. After the extraction process, the liquid phase was separated using 0.45 mm paper filters. The collected water extracts underwent a comprehensive analysis to quantify the presence of leachable metals and other contamination indicators, including barium (Ba), cadmium (Cd), chromium (Cr), copper (Cu), mercury (Hg), molybdenum (Mo), nickel (Ni), lead (Pb), zinc (Zn), selenium (Se), antimony (Sb), arsenic (As), as well as chlorides (Cl), fluorides (F), sulfates (S), dissolved organic carbon (DOC), and total dissolved solids (TDS). The analysis of Zn, Cu, Ba, Pb, Ni, Mo, Cr, and Cd was performed using the ICP-OES technique with a Perkin Elmer Optima 7300 DV instrument, following the PN-EN ISO 11885:2009 standard [46]. The determination of As, Se, and Sb was conducted via ICP-MS using a PerkinElmer NexION 2000 instrument (PerkinElmer, Waltham, MA, USA), following the PN-EN ISO 17294-2:2016-11 standard [47]. Mercury (Hg) was analyzed using the AAS-CV technique with a Perkin Elmer PinAAcle 900T instrument (PerkinElmer, Waltham, MA, USA).

The analysis of volatile organic compounds (VOCs) released from the polyurethane adhesives was performed using a gas chromatography–mass spectrometry (GC-MS) system Agilent GC 7890B/MS 5977B (Agilent Technologies, Santa Clara, CA, USA) equipped with a DB-5MS column (30 m × 0.25 mm × 0.25 μm; Agilent Technologies, Santa Clara, CA, USA). Headspace sampling was carried out using an HS syringe (140 °C) integrated with the MPS Robotic multi-functional autosampler (Gerstel GmbH & Co. KG, Mülheim an der Ruhr, Germany). Approximately 200 mg of sample was sealed in 20 mL vials with PTFE-sealed septa and incubated at 120 °C for 60 min. A volume of 2500 μL of the headspace phase was then injected into the GC inlet. Instrument settings included the following: injector temperature 250 °C; split ratio 10:1; and helium carrier gas at 1.0 mL·min^−1^. The oven program began at 50 °C (1 min hold), ramping to 240 °C at 10 °C·min^−1^. The temperatures of the quadrupole, ion source, and transfer line were set to 150 °C, 230 °C, and 250 °C, respectively. Compound identification was based on comparison with the NIST17 mass spectral library.

Component A consisted of 46% plant-based fatty acid triglycerides, 1% short-chain polyol (ethane-1,2-diol), 50% filler, 0.5% defoamer, and 2.5% drying agent. The technological process included 2 h of drying (polyols and filler) under vacuum (50 bar) at 65 °C. Subsequently, the defoamer and drying agent were added, and the mixture was stirred for 15 min under the same conditions.

The synthesis was carried out using a PC Laborsystem planetary disperser (PC Laborsystem GmbH, Gähwil, Switzerland). Polymeric methylene diphenyl diisocyanate (p-MDI) was used as the isocyanate. The mixing ratio of the components was 100:20, with a reaction index of 1.07.

In the study, the 50% filler was replaced with various combinations of chalk and fly ash, such as:100% chalk—FA0;100% fly ash instead of chalk—FA100;75% fly ash and 25% chalk—FA75;50% fly ash and 50% chalk—FA50;25% fly ash and 75% chalk—FA25;10% fly ash and 90% chalk—FA10.

The prepared compositions were further analyzed to evaluate their impact on the polyurethane adhesive’s physicochemical and mechanical properties.

Additional detailed data are available in the Appendix A.

## 3. Results and Discussion

### 3.1. The Influence of Filler on the Viscosity of 2C PU

The viscosity of Component A was measured in the temperature range of 10 °C to 70 °C for different proportions of chalk and fly ash. The 100% chalk formulation served as a reference for assessing the rheological effects of fly ash replacement.

Viscosity (Figure 1) markedly decreased as temperature increased. At 10 °C, the 100% chalk formulation had a viscosity of 11,430 mPas. For the 100% fly ash variant, the viscosity increased to 19,590 mPas—a 71% rise. A similar trend was observed for intermediate formulations, where viscosity increased proportionally to the fly ash content. At 23 °C, the viscosity dropped to 3638 mPas for chalk and 4919 mPas for fly ash—a 35% increase compared to the reference. In the higher temperature range (30–50 °C), viscosity continued to decrease, but the differences between formulations became smaller. For example, at 50 °C, viscosity was 680 mPas for chalk and 893 mPas for fly ash—a 31% difference. At 60–70 °C, the influence of the filler became minimal, suggesting that temperature was the main factor affecting viscosity. When comparing Component A alone to the A+B mixture, the addition of isocyanate reduced viscosity at all temperatures (Figure 2).

At 10 °C, viscosity was on average 33% lower, and at 50 °C the reduction reached 55%. This confirms that the isocyanate improves the flowability of the system, which is consistent with findings reported by Petrovic et al. [48], who showed that higher free isocyanate content leads to lower viscosity in polyurethane prepolymers. However, the influence of the filler is still visible, especially at lower temperatures.

It is also important to note that, from a technical point of view, it is beneficial when both components of a 2K system have similar viscosity. According to the literature, matching the viscosity of Component A and Component B improves mixing efficiency, leads to more homogeneous curing, and results in better mechanical properties. Differences in viscosity may cause poor mixing and reduce the performance of the adhesive [5,6]. From a practical perspective, the higher viscosity caused by fly ash can make processing more difficult, especially at low temperatures. It can reduce flow, complicate mixing, and require more energy for application. On the other hand, increased viscosity may be useful in applications that need better gap-filling or reduced adhesive run-off. The higher viscosity of fly ash-based formulations is probably caused by particle morphology. Fly ash particles are smaller and more irregular, which creates more internal resistance to flow [49]. Chalk, in contrast, has a more uniform and round shape, which improves dispersion and lowers viscosity [50]. A review of the current literature indicates that the direct effect of fly ash or chalk on the viscosity of two-component polyurethane systems has not yet been systematically studied. The most relevant data come from studies on rigid PU foams, where fly ash was shown to affect reaction time and foam structure. For example, Kuźnia et al. observed that fly ash delayed the foaming reaction, which was probably related to increased viscosity, although it was not measured directly [20,22].

In conclusion, fly ash increases viscosity, which can be both positive and negative depending on the application. It may improve strength and stability but can also reduce workability. Chalk, in contrast, lowers viscosity and improves flow, which is helpful in easy application and spreading. The ability to control viscosity by adjusting the type and amount of filler is a valuable tool in designing polyurethane adhesive formulations tailored to specific industrial needs.

### 3.2. The Influence of Filler Type on the Physicochemical and Mechanical Properties of 2C PU

Following the analysis of the impact of filler type on the rheological properties of polyurethane formulations, the next stage of the study focused on assessing the effect of replacing chalk with fly ash on the density, thermal conductivity, hardness, and calorific value of polyurethane adhesives (Table 1).

The results showed that the density of the adhesive formulation decreases with increasing fly ash content. The formulation containing 100% chalk had the highest density (1.46 g/cm^3^), while the formulation with 100% fly ash reached 1.32 g/cm^3^. Intermediate formulations showed density values ranging from 1.39 to 1.45 g/cm^3^, confirming the gradual influence of the filler on the density of the system. This effect may result from differences in particle morphology and their ability to pack efficiently within the polyurethane matrix. While few studies have directly examined this effect in polyurethane adhesives, related research on polyurea-based composites supports this observation. Qiao and Wu found that adding 30 vol% fly ash particles to a polyurea matrix reduced composite density by about 12%, due to the low bulk density and hollow, porous structure of the filler [51]. These findings suggest that fly ash can similarly reduce density in other polymer systems, including PU adhesives. Calcium carbonate (CaCO_3_) is commonly used as a filler in polyurethane foams. Usman et al. demonstrated that increasing its content up to 30 wt% led to a gradual increase in foam density, with a particularly sharp rise observed above 20%. This effect was attributed to the ability of filler particles to occupy space within the matrix and reduce the material’s porosity [52]. Whether lower density is beneficial or not depends on the application. In some cases, reduced density may be an advantage—for example, to reduce material weight or improve logistics. However, it can also require adjustments in dosing and application parameters, especially in automated systems. Lower density can also affect packaging and logistics. Less dense adhesives occupy more volume for the same weight, which may require larger packaging or adjustments in filling equipment. In systems with volume-based dosing (e.g., cartridges, pouches), this could result in lower net weight per unit. On the other hand, a lower overall mass may reduce transport costs, which is beneficial in large-scale production. For this reason, filler selection should consider not only processing and performance but also packaging efficiency and supply chain impact. The results indicate that density decreases with the increasing proportion of fly ash. A decrease in density may affect the stability of the adhesive during application—potentially improving it in some cases but also requiring adjustments in dosing parameters. This is particularly important in industrial systems, where precise control over viscosity and the amount of adhesive applied is crucial for joint quality.

Another key parameter was the thermal conductivity of the adhesive, which increased with a higher proportion of fly ash. In the 100% chalk formulation, the thermal conductivity was 0.296 W/m.K, while in 100% fly ash, it rose to 0.3936 W/m.K. This increase is likely due to the different crystalline structure of fly ash and its greater ability to conduct heat compared to chalk, which serves as an insulator. This trend is consistent with previous studies on rigid polyurethane foams. Zygmunt-Kowalska et al. reported that adding 10 wt% fly ash improved the thermal insulation properties of PU foam, confirming the effect of filler type on thermal conductivity [53]. Intermediate formulations confirmed a gradual increase in thermal conductivity, indicating that this property can be controlled through appropriate filler ratio selection. From an application perspective, increased thermal conductivity can be beneficial for adhesives used in bonding metal components, where local heating of the joint may affect mechanical properties. Conversely, formulations based on chalk may be preferred in construction or automotive applications, where limiting heat loss within composite structures is a priority [22,54,55].

The high heating value is an important parameter in the context of the fire safety of polyurethane materials. The tests conducted showed that replacing chalk with fly ash led to an increase in this parameter. In the 100% chalk formulation, the heating value was 18,747 J/g, whereas in the formulation with 100% fly ash, it reached 21,659 J/g. This increase can be attributed to differences in the chemical composition of the fillers, with fly ash potentially containing a higher proportion of organic substances that contribute to combustion heat. Similar effects were observed by Li et al., who reported that the addition of fly ash can influence the thermal degradation rate and improve the fire resistance of polyurethane composites [22]. Furthermore, patent studies show that reducing the heating value of polyurethane adhesives (to below 20 MJ/kg) may be necessary to meet fire resistance requirements in construction applications [56]. Therefore, the increased heating value resulting from fly ash addition may require the use of flame retardants, especially in polyurethane adhesives used in buildings, where strict fire safety regulations must be met. As Li et al. emphasize, improving the thermal and fire resistance of polyurethane systems is not only of theoretical interest but also of high practical relevance [57].

Hardness analysis using the Shore A scale showed that increasing the fly ash content leads to a reduction in hardness, indicating greater flexibility of the adhesive. The 100% chalk formulation reached 85 Shore A, while the 100% fly ash formulation had a hardness of 73 Shore A. Intermediate formulations exhibited hardness values between 76 and 80 Shore A, suggesting that the mechanical properties of the adhesive can be tailored based on the desired final joint hardness. This trend is consistent with previous studies. Li et al. demonstrated that the addition of fly ash disrupts polymer chain interactions and increases chain flexibility in polyurethane composites, resulting in softer and more elastic materials [22]. On the other hand, calcium carbonate has been shown to increase Shore A hardness and improve dimensional stability. According to Hussain et al., adding just 5 phr of precipitated calcium carbonate (PCC) significantly increased the hardness and thermal stability of PU-based materials [58]. Harder formulations are more resistant to mechanical deformation, making them suitable for applications requiring stiffness and dimensional stability. Conversely, adhesives with more fly ash may offer better stress compensation in dynamic applications, such as joints exposed to varying thermal or mechanical conditions.

Tensile strength tests (Table 2) show that both maximum load and stress values remained consistent across all formulations, regardless of the chalk-to-fly ash ratio. For galvanized steel + XPS bonds, the maximum load values ranged from 1140 N to 1187 N, while stress at maximum load varied between 429 kPa and 460 kPa. For aluminum + XPS bonds, the maximum load values ranged from 1100 N to 1189 N, and the corresponding stress values ranged from 429 kPa to 496 kPa. Bond strength values varied slightly between formulations, with no obvious dependency on the filler ratio. All tested samples exhibited 100% failure within the XPS material, with no damage observed in the adhesive layer. This confirms that the cohesive strength of the adhesive exceeds the tensile strength of the XPS substrate, indicating good adhesion to both galvanized steel and aluminum, as well as high overall bond quality. It is worth noting that similar conclusions were drawn in studies on polyurethane adhesives used in external thermal insulation composite systems (ETICS) based on mineral wool. Sudoł and Kozikowska demonstrated that the failure of the adhesive joint was determined by the weakest component of the system, in that case, the insulation material itself. Although their research focused on mineral wool, this behavior may be extended to XPS-based systems, where the mechanical properties of the substrate can also act as a limiting factor for overall bond strength [59].

In summary, regardless of the filler composition, the polyurethane adhesive maintained high mechanical performance and provided strong bonding to XPS in combination with both galvanized steel and aluminum. These results indicate that modifying the filler content does not compromise the adhesive’s mechanical integrity and supports its applicability in demanding industrial conditions, particularly in multilayer insulation systems.

The analysis of the influence of different proportions of chalk and fly ash on the mechanical properties (Table 3) of a two-component polyurethane adhesive (2C PU) showed that, at 10 °C and 23 °C, the mechanical strength remained similar regardless of the filler composition. Maximum stress values ranged between 5.29 and 5.63 MPa. However, noticeable differences appeared at elevated temperatures. At 40 °C, the sample with 100% fly ash showed a maximum stress of 4.68 MPa, while compositions with more chalk reached up to 5.31 MPa (at 90% chalk). At 80 °C, the strength dropped further—down to 3.96 MPa for 100% fly ash and 4.81 MPa for 90% chalk. The lowest values were recorded at 120 °C: 3.37 MPa for 100% fly ash and 3.71 MPa for 90% chalk. A similar pattern was observed in the stress at break. At room temperature, the differences between samples were small, but at 120 °C, the lowest value (3.33 MPa) was again recorded for the 100% fly ash formulation. The flexibility of the adhesive, measured as elongation at break, also depended on the type of filler. At 10 °C and 23 °C, all samples showed similar values (60–63%). However, as the temperature increased, flexibility decreased—especially in samples with high fly ash content. At 120 °C, the elongation at break dropped to 33.10% for 100% fly ash, and 40.20% for 90% chalk.

There are no scientific publications available that present the mechanical properties of 2C PU adhesive films filled with chalk or fly ash, tested using ISO 37:2024 [43] at various temperatures. The closest related works describe polyurethane systems tested in different conditions—such as composites, coatings, or adhesives with other fillers. For example, Torro-Palau et al. [60] studied solvent-based polyurethane adhesive films filled with sepiolite, tested according to ISO 37. They reported tensile strengths in the range of a few MPa, which decreased significantly with higher filler content. De Smet et al. [61] tested bio-based 2C PU films for textile coatings using ISO 13934-1 [62] but did not publish specific stress values. Zhai et al. [63] investigated polyurethane–fly ash composites, but focused on flexural strength, not on adhesive films.

Since there are no directly comparable results in the literature, the values obtained in this study should be considered new experimental data for this specific adhesive formulation. The results suggest that chalk helps maintain better mechanical performance at higher temperatures, while fly ash causes a more noticeable reduction in strength and flexibility. Differences between filler types became more pronounced above 40 °C, which is important when designing adhesives for use in elevated temperature conditions. The observed decline in tensile strength and elongation at higher temperatures can be attributed to thermal softening of the polyurethane matrix, which leads to decreased intermolecular forces and reduced entanglement of polymer chains. This reduces the material’s ability to resist deformation and mechanical stress under load, as supported by previous findings on polyurethane systems [64].

### 3.3. The Influence of Filler Type on VOC Release and Metal Leachability from 2C PU

The GC-MS analysis of the tested formulations yielded a limited number of chromatographic signals; however, none of the detected compounds reached a match factor (MF) of ≥700, the minimum threshold for confident identification according to NIST guidelines [65,66]. As a result, no individual VOCs could be reliably identified, and the detected signals were excluded from further compound-specific interpretation.

The most distinct chromatographic profile (Figure 3) was observed for the formulation containing 100% chalk (FA0), possibly indicating low chemical reactivity or high purity of the components. While the analysis was qualitative, the absence of reliably identified VOCs may suggest low emission potential. This contrasts with literature reports, such as the study by Kozicki and Guzik, which identified numerous VOCs in commercial flooring adhesives using the same technique. It is worth noting that the study conducted by Kozicki and Guzik examined a different type of polyurethane adhesive, which may have contained additives or raw materials responsible for the presence of numerous VOCs. This could explain the discrepancy in results despite the use of the same analytical technique [67].

Nevertheless, due to the limitations of qualitative screening, additional quantitative VOC analysis is necessary to verify emission levels and assess conformity with environmental certification criteria such as A+, AgBB, EMICODE^®^, M1^®^, LEED, or BREEAM.

Analysis of aqueous extracts of 2C PU samples showed that all tested formulations—regardless of the chalk-to-fly ash ratio—meet the requirements specified in the Regulation of the Minister of Economy of 16 July 2015, which governs the acceptance of waste for landfilling in sites other than hazardous and inert [68]. The elution rates of toxic elements (Cd, Hg, Pb, Cr, As, Ni) and potentially mobile substances (Sb, Se, Mo, Zn, Cu, Ba) did not exceed the permissible limits (Figure 4). For many components, such as Cd, Pb, Zn, Cu, Ni, and Hg, the concentrations remained below the limit of detection (<LOD), indicating only trace levels in the water extracts. The highest recorded elution rates of molybdenum (0.62 mg/kg) and selenium (0.20 mg/kg)—observed in the FA100 formulation—represented just 6.2% and 40% of their respective regulatory thresholds. Arsenic reached a maximum of 0.33 mg/kg, amounting to 16.5% of the permissible limit. Similarly, the maximum barium concentration (2.0 mg/kg) corresponded to only 2% of the acceptable value (100 mg/kg). These results demonstrate that even in the most extreme formulation—where chalk is fully replaced by fly ash—the material remains compliant with environmental safety standards for commercial use. This is also supported by the publication of Wowkonowicz et al., which emphasizes the safe use of industrial waste materials in the construction sector, provided they meet elution and emission standards for hazardous substances [69]. The elution rates of inorganic anions such as sulfates, fluorides, and chlorides also remained below the thresholds defined in the regulation (Table 4). The highest leachability of sulfates (430 mg/kg) and fluorides (13 mg/kg) was observed in the FA100 sample, representing only 21.5% and 8.7% of the acceptable limits (2000 mg/kg and 150 mg/kg, respectively). The chloride elution in all samples remained below 50 mg/kg—which is only 3.3% of the regulatory limit (1500 mg/kg). Another important environmental parameter is the dissolved organic carbon (DOC) and total dissolved solids (TDS), which also remained at low levels (up to 4260 mg/kg TDS and 340 mg/kg DOC), corresponding to less than 7% of their maximum permitted leachability. From an industrial perspective, these results are highly relevant. They confirm that polyurethane adhesives containing fly ash do not release harmful substances into the environment—either during application or under service conditions. This means that the adhesive can be safely used to bond construction materials such as galvanized steel, aluminum, or XPS boards, without the risk of surface degradation, galvanic corrosion, or chemical damage. In addition, the absence of aggressive ions (Cl^−^, F^−^, SO_4_^2−^) protects components against potential corrosion effects, which is especially important in humid environments and enclosed spaces.

### 3.4. Does the Fly Ash Cease to Be Waste?

The results of this study raise an important question in light of the EU Waste Framework Directive (2008/98/EC) [70]: Can the fly ash used in the adhesive formulation meet the criteria for end-of-waste status and be considered a recycled material?

According to Article 6 of the directive, material may cease to be classified as waste if it has undergone a recovery operation, including recycling, and meets all of the following criteria:

#### 3.4.1. The Substance or Object Is Commonly Used for Specific Purposes

In this study, fly ash was used as a functional filler in two-component polyurethane adhesives, replacing chalk in varying proportions. Its application resulted in favorable physicochemical and mechanical properties, confirming its suitability for a defined industrial function.

#### 3.4.2. A Market or Demand Exists for Such a Substance or Object

Fly ash is already widely used in construction materials, such as cement and concrete. This study expands its application by demonstrating its successful use in adhesives. Given the increasing emphasis on circular economy practices, market demand for such uses is present and expected to grow.

#### 3.4.3. The Substance or Object Fulfills the Technical Requirements for Specific Purposes and Meets the Existing Legislation and Standards Applicable to Products

The fly ash-based adhesives met the necessary mechanical strength, durability, and thermal stability criteria for construction-grade adhesives. Furthermore, they complied with environmental standards—all measured contaminant leaching levels were significantly below regulatory thresholds, including molybdenum (6.2% of the limit) and selenium (40%).

#### 3.4.4. The Use of the Substance or Object Will Not Lead to Overall Adverse Environmental or Human Health Impacts

Based on the leachability data and preliminary VOC findings, the use of fly ash in adhesive formulations does not pose a significant risk to human health or the environment. This supports its safe application in construction, including indoor use.

Given that the fly ash underwent a functional transformation from industrial waste into a value-added secondary raw material, the study clearly demonstrates the characteristics of material recovery and recycling. Its use in adhesive systems, validated through laboratory-scale production and performance testing, confirms a new, well-defined function without environmental burden. Therefore, under the Waste Framework Directive, the tested fly ash fulfills the criteria for end-of-waste status. It should no longer be classified as waste but as a recycled material, contributing to circular resource flows and reducing reliance on virgin fillers such as chalk.

## 4. Conclusions

The findings of this study demonstrate that fly ash can serve as an effective partial or full replacement for chalk in two-component polyurethane adhesives, maintaining both application performance and regulatory compliance. All tested formulations met the leachability thresholds defined by the Regulation of the Minister of Economy, with the highest observed concentrations of molybdenum (0.62 mg/kg) and selenium (0.20 mg/kg) remaining well below permissible levels. Elution values for dissolved organic carbon (DOC) and total dissolved solids (TDS) were also low, not exceeding 340 mg/kg and 4260 mg/kg, respectively. Preliminary GC-MS screening did not allow for the confident identification of specific volatile organic compounds (VOCs), as all detected substances showed match factors below the threshold of 700 recommended by NIST. Although this may indicate limited VOC presence, the conclusion remains tentative and requires confirmation through further quantitative analysis. Mechanical testing showed cohesive failure within the adherend material, confirming that the bond strength of the adhesives exceeded the internal strength of the substrates, which supports the high interfacial performance of the fly ash-modified polyurethane formulations when applied to galvanized steel, aluminum, and XPS.

Moreover, fly ash-based adhesives not only meet environmental requirements but also maintain the functional performance necessary for construction applications, underlining their potential for sustainable materials technology. Based on the criteria defined in the EU Waste Framework Directive, the tested fly ash can be considered a recycled material that has lost its waste status. These conclusions point to strong application potential; however, the scope of the study was limited to short-term mechanical and chemical testing. Long-term durability under conditions such as variable humidity, temperature changes, cyclic loading, and fire exposure was not assessed. Additionally, the study did not include structural and morphological characterization methods (e.g., SEM, FT-IR, or XRD), which limits understanding of filler dispersion and interaction within the polyurethane matrix. Given the limitations of the qualitative VOC screening method, further quantitative GC-MS analysis is essential to accurately assess emissions and confirm compatibility with environmental certification systems such as A+, AgBB, EMICODE^®^, M1^®^, LEED, or BREEAM.

## Figures and Tables

**Figure 1 materials-18-03591-f001:**
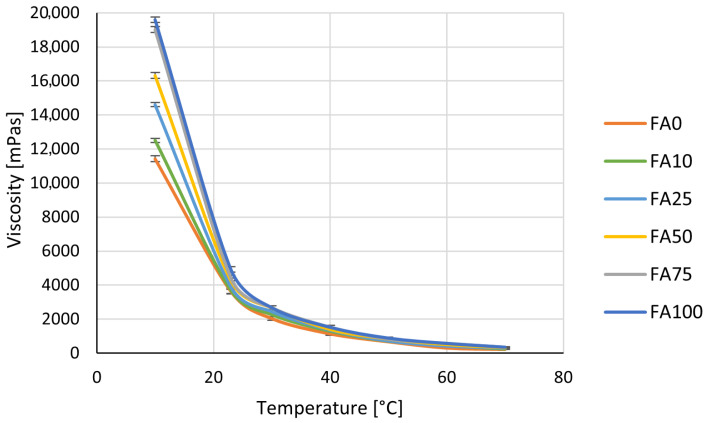
Viscosity of Component A in polyurethane adhesive as a function of temperature and filler composition.

**Figure 2 materials-18-03591-f002:**
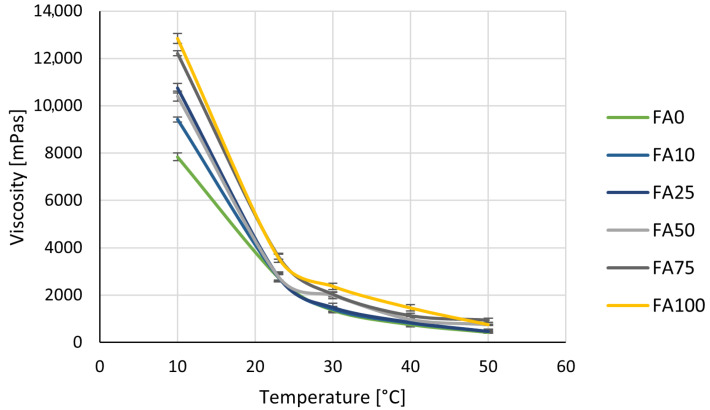
The viscosity of the Component A + Component B mixture in polyurethane adhesive as a function of temperature and filler composition.

**Figure 3 materials-18-03591-f003:**
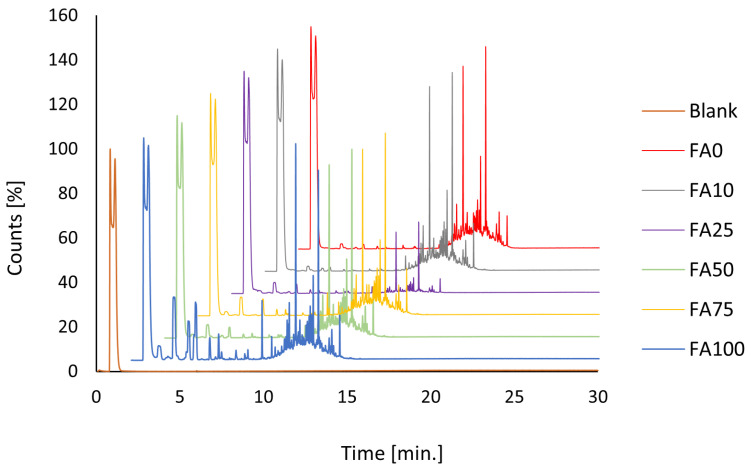
Comparison of GC-MS profiles for formulations with varying chalk-to-FA ratios.

**Figure 4 materials-18-03591-f004:**
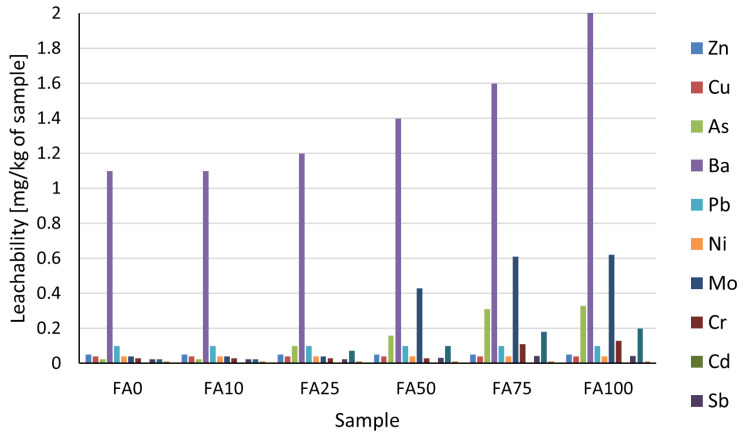
The pollutant elution rate (elements) from composite formulations (mg/kg). Most of the elements were detected below quantification limits and remained well within regulatory thresholds.

**Table 1 materials-18-03591-t001:** Influence of chalk and fly ash content on the physicochemical and mechanical properties of two-component polyurethane adhesives.

Parameter	FA0	FA10	FA25	FA50	FA75	FA100
Density [g/cm^3^]	1.46 ± 0.01	1.45 ± 0.01	1.44 ± 0.01	1.42 ± 0.01	1.39 ± 0.01	1.32 ± 0.01
Open time [min]	70 ± 0	70 ± 0	70 ± 0	70 ± 0	70 ± 0	70 ± 0
Thermal conductivity [W/m.K]	0.2960 ± 0.0021	0.3277 ± 0.0010	0.3355 ± 0.0030	0.3361 ± 0.0017	0.3814 ± 0.0019	0.3936 ± 0.0027
High heating value [J/g]	18,700 ± 200	18,700 ± 300	19,300 ± 200	20,000 ± 300	20,400 ± 300	21,700 ± 200
Shore hardness A	85 ± 2	80 ± 3	78 ± 3	78 ± 3	76 ± 2	73 ± 4

**Table 2 materials-18-03591-t002:** Influence of chalk and fly ash content on the tensile strength perpendicular to faces. All samples exhibited 100% failure in XPS.

Parameter: Tensile Strength Perpendicular to Faces		FA0	FA10	FA25	FA50	FA75	FA100
galvanized steel + XPS	Maximum Load [N]	1150 ± 30	1140 ± 30	1180 ± 60	1160 ± 70	1190 ± 30	1160 ± 40
	Stress at Maximum Load [kPa]	460 ± 20	440 ± 60	430 ± 40	460 ± 40	450 ± 20	440 ± 30
aluminum + XPS	Maximum Load [N]	1160 ± 30	1150 ± 30	1180 ± 70	1170 ± 40	1170 ± 40	1100 ± 70
	Stress at Maximum Load [kPa]	460 ± 30	500 ± 30	440 ± 40	430 ± 30	460 ± 50	460 ± 40

**Table 3 materials-18-03591-t003:** Influence of chalk and fly ash content on the tensile stress–strain properties.

PARAMETER	TEMP.	FA0	FA10	FA25	FA50	FA75	FA100
Maximum Stress [MPa]	T = 10°	5.61 ± 0.11	5.60 ± 0.24	5.59 ± 0.20	5.50 ± 0.15	5.42 ± 0.21	5.36 ± 0.06
	T = 23°	5.63 ± 0.16	5.61 ± 0.20	5.55 ± 0.19	5.53 ± 0.22	5.40 ± 0.08	5.29 ± 0.10
	T = 40°	5.37 ± 0.12	5.31 ± 0.16	5.29 ± 0.15	4.90 ± 0.19	4.86 ± 0.20	4.68 ± 0.12
	T = 80°	4.85 ± 0.10	4.81 ± 0.17	4.81 ± 0.16	4.76 ± 0.20	4.43 ± 0.24	3.96 ± 0.16
	T = 120°	3.86 ± 0.01	3.71 ± 0.2	3.58 ± 0.10	3.53 ± 0.18	3.43 ± 0.14	3.37 ± 0.20
Stress at Break [MPa]	T = 10°	4.67 ± 0.10	4.67 ± 0.21	4.57 ± 0.14	4.56 ± 0.24	4.48 ± 0.21	4.47 ± 0.14
	T = 23°	4.77 ± 0.22	4.63 ± 0.20	4.57 ± 0.17	4.55 ± 0.17	4.48 ± 0.20	4.48 ± 0.16
	T = 40°	4.68 ± 0.19	4.50 ± 0.19	4.30 ± 0.18	3.99 ± 0.18	3.98 ± 0.18	3.94 ± 0.18
	T = 80°	4.58 ± 0.17	4.50 ± 0.16	4.29 ± 0.24	3.94 ± 0.20	3.83 ± 0.09	3.81 ± 0.19
	T = 120°	3.56 ± 0.20	3.51 ± 0.13	3.51 ± 0.17	3.46 ± 0.21	3.33 ± 0.05	3.35 ± 0.20
Elongation at Break [%]	T = 10°	63.20 ± 0.13	62.80 ± 0.14	62.50 ± 0.14	62.30 ± 0.13	60.90 ± 0.16	59.99 ± 0.25
	T = 23°	62.20 ± 0.14	61.23 ± 0.13	61.10 ± 0.13	61.80 ± 0.14	60.60 ± 0.14	59.58 ± 0.11
	T = 40°	61.90 ± 0.15	61.30 ± 0.20	60.91 ± 0.15	60.70 ± 0.21	59.40 ± 0.20	58.10 ± 0.11
	T = 80°	44.90 ± 0.16	43.60 ± 0.13	43.10 ± 0.16	42.10 ± 0.20	42.90 ± 0.16	42.10 ± 0.13
	T = 120°	40.60 ± 0.08	40.20 ± 0.16	40.10 ± 0.17	39.60 ± 0.15	38.30 ± 0.18	33.10 ± 0.11

**Table 4 materials-18-03591-t004:** The results of other pollutant elution rate (mg/kg) from 2C PU [mg/kg s.m.].

Parameter	DOC	TDS	Sulfates	Fluorides	Chlorides
Unit	mg/kg s.m.				
FA0	340	3200	<100	<1.0	<50
FA10	310	2360	<100	<1.0	<50
FA25	320	2460	150	4.7	<50
FA50	200	3220	250	7.0	<50
FA75	220	3780	400	11	<50
FA100	240	4260	430	13	<50
acceptable limit	800	60,000	2000	150	1500

## Data Availability

The original contributions presented in this study are included in the article/Appendix A. Further inquiries can be directed to the corresponding authors.

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
