# Peer review of "The Impact of Substituting Chalk with Fly Ash in Formulating a Two-Component Polyurethane Adhesive on Its Physicochemical and Mechanical Properties"

_materials, 2025, doi:10.3390/ma18153591_

Round 1

Reviewer 1 Report

Comments and Suggestions for Authors

Dear authors,

a very clearly structured and detailed contribution, thank you very much. I have only a few comments: Please update the standards used. Do not refer to standards that have been out of date for some time and have been revised. For example, EN ISO 10368:1995 is now EN ISO 10368:2022, so the fracture pattern you are evaluating is the cohesive failure in the substrate CSF and is therefore described even more precisely than in the withdrawn version.
Furthermore, the contents of line 287-291 are duplicated with line 311-315. The statement is the same. Please adjust this.

Reviewer 2 Report

Comments and Suggestions for Authors

The author presents a study investigating the substitution of chalk with fly ash as a filler in two-component polyurethane (2K PU) adhesives. The authors aim to assess the physicochemical, mechanical, and environmental implications of this replacement, highlighting its potential to support circular economy principles. Various formulations with different chalk/fly ash ratios were tested for viscosity, thermal conductivity, mechanical strength, hardness, and environmental safety (VOC emissions and leachability). However, there are a few scientific issues that should be addressed to strengthen the manuscript's credibility before its acceptance.

  1. In the introduction, the author claims that the use of fly ash in 2K PU adhesives is "insufficiently explored," but does not differentiate adhesives from other PU systems like foams and composites. Clearly state the novelty in the context of adhesives and provide more specific literature comparisons.
  2. The authors provide results of mechanical and thermal tests with standard deviations. However, no statistical significance testing (e.g., ANOVA, p-values) is used to support statements like “no significant differences.”
  3. The analysis of VOCs using GC-MS is qualitative, and the conclusion of “low emission potential” is based on signal match factors alone. This limitation is acknowledged, but it would strengthen the study to include preliminary quantitative data or retention times with concentration thresholds where possible.
  4. The authors study evaluates only short-term mechanical and thermal performance. while acknowledged in the conclusion, it should be emphasized earlier that long-term performance (aging, humidity cycling, fire resistance) is not covered and should be addressed in future work.
  5. The Introduction includes a long, nearly exhaustive list of waste-based PU additives (e.g., coconut fibers, fish collagen, hazelnut shells), which dilutes focus. Shorten this section to highlight only the most relevant analogs (e.g., fly ash in rigid foams or composites) and clarify how this research builds upon or differs from them.
  6. Add statistical tests to validate claims of "no significant differences."
  7. Use a standardized terminology (2C PU vs. 2K PU) throughout the manuscript.
  8. Figures (especially viscosity plots and GC-MS outputs) are poorly designed: No error bars, and even legends are not standardized.
  1. Viscosity and stress-strain plots are clearly Excel exports or image with compressed resolution. Axes lack precision (e.g., unclear units, ambiguous labels).
  2. Replot all graphs using vector-based plotting tools (e.g., Origin, Python/matplotlib, R。
  3. GC-MS results are based solely on match factors <700, which the authors interpret as low emissions. This is not a valid conclusion. The absence of identified peaks does not equate to low emission, just low spectral matching. Perform quantitative VOC testing, or at a minimum, report total ion count (TIC) comparisons and potential peak areas. Otherwise, remove any conclusive claims regarding VOC compliance or emission safety.
  4. Why does mechanical strength and stress decline in specimens with increasing temperature? Reason and provide Ref。
  5. In the manuscript, the author makes repeated claims about the particle morphology, crystallinity, and interaction of chalk vs. fly ash with the polyurethane matrix.  However, no structural characterization methods such as X-ray Diffraction (XRD), SEM, or FT-IR are included to support claims about particle shape and dispersion. To identify and compare mineral phases or crystalline structures and confirm chemical bonding or interaction between filler and matrix, provide this analysis too.

Reviewer 3 Report

Comments and Suggestions for Authors

The paper is devoted to the estimation of the effect of replacing chalk with fly ash in a two-component polyurethane adhesive on its properties. It was found that this substitution did not significantly alter the properties of the adhesive while allowing for the use of an inexpensive industrial waste product instead of chalk. The paper is suitable for publication in Materials after minor revision taking into account the following comments.

  • The abbreviations 2C PU, 2K PU were introduced several times in the manuscript.
  • The abbreviations for the adhesives are too long. I would propose shortening them to FA0, FA25 etc.
  • It would be better to use the following order of samples in the tables (including those below the Figures) and some Figures: FA0, FA25, …, FA100. Otherwise, it is difficult to see the effect of ash on the properties.
  • Tables 1,2: If the error is relatively high, the last few digits may not be significant. For instance, 21659±232 is better to write as 21700±200.

Comments on the Quality of English Language

I am not qualified enough to judge about the quality of English language

Reviewer 4 Report

Comments and Suggestions for Authors

Dear authors,

The comments are found in the attached file.

best regards

Comments on the Quality of English Language

The languague can be improved.

Round 2

Reviewer 4 Report

Comments and Suggestions for Authors

Accept